# Roles of Keratins in Intestine

**DOI:** 10.3390/ijms23148051

**Published:** 2022-07-21

**Authors:** Jeongwon Mun, Whan Hur, Nam-On Ku

**Affiliations:** 1Department of Bio-Convergence ISED, Underwood International College, Yonsei University, Seoul 03722, Korea; jamiemun@yonsei.ac.kr (J.M.); hur.whan@yonsei.ac.kr (W.H.); 2Interdisciplinary Program of Integrated OMICS for Biomedical Sciences, Yonsei University, Seoul 03722, Korea

**Keywords:** keratin, intestine, IBD, CRC, transgenic mice

## Abstract

Keratins make up a major portion of epithelial intermediate filament proteins. The widely diverse keratins are found in both the small and large intestines. The human intestine mainly expresses keratins 8, 18, 19, and 20. Many of the common roles of keratins are for the integrity and stability of the epithelial cells. The keratins also protect the cells and tissue from stress and are biomarkers for some diseases in the organs. Although an increasing number of studies have been performed regarding keratins, the roles of keratin in the intestine have not yet been fully understood. This review focuses on discussing the roles of keratins in the intestine. Diverse studies utilizing mouse models and samples from patients with intestinal diseases in the search for the association of keratin in intestinal diseases have been summarized.

## 1. Introduction

Keratins are intermediate filament proteins that are expressed in epithelial cells. They are mainly responsible for maintaining the shape of the cell and protecting it from stress, both mechanically and nonmechanically [1,2]. The simple epithelial keratins (SEKs) are divided into two types: type I and type II. Type I is the acidic type (K9-K28 and K31-K40), and type II is the basic type (K1-K8 and K71-K86) [2,3,4]. With at least one of each type of SEK, an obligate noncovalent heteropolymer is assembled, and the structure includes an α-helical rod in the center of the structure and a non-α-helical head and tail at the ends of the structure [5,6]. The coiled-coil α-helical rod consists of subdomains of coils: 1A, 1B, 2A, and 2B. Each of these subdomains is connected by linkers that consist of L1, L2, and L12. Unlike the rod, the head and tail are less conserved and show variations [5,7]. Out of these, K18, K19, and K20 from acidic type I and K7 and K8 from basic type II are found in the intestines [8,9]. The undifferentiated cells of the crypts express K18, and the villus cells significantly express K20 [8]. K8 and K19 are greatly found in the epithelium linings of both the small and large intestines [8]. As for K7, this is mostly found in the simple epithelia and basal cells [10].

Human patients, mice models, and cell cultures have been consistently used to research the diverse functions of keratins, but the defined roles in intestine are not yet fully understood. Even though the pattern of keratin expression is slightly different in humans and mice (Table 1), the findings in mouse models can be utilized to study keratin-associated human intestinal disease. The roles of keratins can be deeply understood using knock-out or transgenic mouse models that have induced colon injury. This review concentrates on keratins associated with: (i) the distribution of keratins in intestine, (ii) intestine phenotypes of keratin-related mouse models, and (iii) keratin mutations associated with intestine diseases.

## 2. Distribution of Keratins in Intestine

### 2.1. Keratin Expression in the Small Intestine and Colon

A single layer of epithelial cells is folded in a specific way to form villi and crypts, which make up the small intestine. The large intestine, however, only has crypts. In the villi, the differentiated epithelial cells are accumulated. These cells do not divide and are composed of enterocytes, goblet cells, and enteroendocrine cells [19]. On the other hand, the crypts contain undifferentiated cells, of which some, except Paneth cells, will eventually migrate toward the villi in the small intestine [19]. Although the cells and compartments are different between the small and large intestine, keratin expression based on different cell types of the intestine has not yet been observed. However, the overall keratin patterns of the small intestine and colon show a similar framework.

The main keratins found in the normal intestinal epithelial cells (IECs) are K8, K18, and K19 [15]. These keratin distributions throughout each cell are not even. The keratin pairings have been shown by way of in vitro assays, but many of these pairings have not been shown to exist in cells. Nevertheless, K8, along with its partners K18 and K19, has been observed in many studies of normal IECs in humans and mice [9,20]. On the other hand, K19 also has a partner, K7, but only K19 has been frequently observed in the normal cells [21]. The other keratins, K7 and K20, have been identified, but the role of these keratins varies depending on the study. Some studies consider both K7 and K20 as being present in the normal IECs [11,22]. However, other studies disagree on the presence of K7, because the expression is either too small to have a significant value or is essentially absent [23].

Nonetheless, the atypical expressions of K7, K8, and K20 have been found to have potential roles in abnormal intestinal cells [13,24]. For instance, the strong expression of K20 has been linked to colorectal cancers (CRCs). In comparison to normal cells, the expression of K20 in CRC has been shown to be extremely strong with an absence of K7 or with a weak expression of K7 [16,17,25,26]. Furthermore, mutations in K8 have been found in inflammatory bowel disease (IBD) patients [14]. However, the molecular mechanism of how the K8 mutation is associated with IBD remains to be determined. Overall, the distribution of keratins in the intestine are summarized in Table 1, based on normal and abnormal intestinal cells. The table also illustrates the differences between human and mouse normal intestinal cells.

### 2.2. Differences of Keratin Expression in Human and Mouse Intestine

The expression of keratin is different in human and mouse intestines. It further differs in the small and large intestines of both species. For the keratin in normal human small intestine, the presence of K8, K18, and K19 is observed in the small intestine [12]. K20 is also present, but it is expressed strongly in the villus-lining epithelia [27]. On the other hand, the presence of all four keratins, K8, K18, K19, and K20, is found in the human colon [28]. In a study observing the simple epithelial keratins, the keratins within the human colon were studied through immunohistochemistry [28]. The single-layered epithelium in the colon contained an even expression of K8, K18, and K19 [28]. However, K20 was found in luminal cells rather than the crypt cells [28].

As for the mice, the distribution of keratin also slightly differs between the intestine and its area. For the small intestine, K8 and K19 are evenly distributed within the crypt–villus axis [11]. However, K7 and K18 are found mostly in the crypt base, while K20 is found mostly in the villus tip [8]. Furthermore, the mouse colon contains a balanced dispersion of K8, K18, and K19 in both the crypt base and villus tip [11]. K7 is found mostly near the villus tip, and K20 is found mostly near the crypt base [8]. In addition, it was found that K20 is more plentiful in the small intestine than the colon [11].

Keratins have many different functions and are found in various locations of different organs and organisms. The reasons for the differences in location of K7, K8, K18, K19, and K20 for both human and mice remain unclear, and the defined attribution of a particular function to each of the keratins is ambiguous. In addition, some studies even go further to show that K23 and K80 are also weakly expressed in the colon [28,29]. Nonetheless, more experimental data are needed to clearly identify the main cause of these differences in keratin location of both human intestine and mice intestine, as well as the specific function.

### 2.3. General Roles of Keratin

It is known that keratins play crucial roles in stress protection, cell integrity and structure, and protein targeting [22,30,31]. Similar to how the functions of keratin differ depending on its location within an organ, the functions of keratin also vary depending on the organs. Keratins are responsible for protecting the liver from nonmechanical stress [32], and they are involved in maintaining the intestinal barrier by their association with membrane junctions [9]. The keratin filament networks expand through the cytoplasm and attach to intercellular junctions such as the lateral desmosome and basal hemidesmosome of colonocytes, which allowing keratin linkage to cell–cell and cell–matrix adhesions [9]. As a result, keratins have crucial roles for maintaining intercellular adhesion, tissue morphology, and membrane stability. If there is a loss of keratin, there could potentially be weaker cell–cell and cell–matrix contacts.

Despite the well-identified associations between the crucial roles and the type of keratin in the liver, the functional roles of keratin in both the small and large intestines have not yet been fully understood. However, studies regarding mouse models have provided steps closer to finding the roles. For instance, studies regarding K8 null mouse models have provided potential roles of keratin in stress protection, inflammation, and cell proliferation [33,34]. Nonetheless, the roles of keratin in intestine and their associations with intestinal diseases are still being vigorously studied using cell culture systems, mice, and samples from patients with intestinal diseases.

## 3. Intestinal Phenotypes of Keratin-Related Mouse Models

### 3.1. DDS and AOM Murine Models

Chemicals used to induce mouse models to colon injury usually come in two forms: dextran sulfate sodium (DSS) and azoxymethane (AOM). These animal models allow for clearer explanations on the pathogenic mechanisms of inflammatory bowel diseases and colorectal cancer. Dextran sulfate sodium (DSS) is a chemical with anticoagulant properties and is used to predispose colon injury in mouse models. The DSS model is frequently used in inflammatory bowel disease, because it is easy, straightforward, and quick [35]. The normal concentration of DSS that is used is around 2–2.5%, but depending on the mouse model and study, 5% can also be used [11,22]. Through this, colitis induced in mice is the most comparable to ulcerative colitis found in humans [36].

On the other hand, azoxymethane (AOM) is a carcinogen that is intraperitoneally administered into mouse models [37]. With this, the mice are induced to damage of the colonial mucosal, neoplasia, and other precursors of colon cancer [38]. Through this stress induction, the mouse goes through stages similar to the stages of human colon cancer. The mice start off at the aberrant crypt foci (ACF) stage or the stage where the precursors of colon cancer start [39]. Then, they develop adenoma and, finally, carcinoma. Even though these mouse models may be a good indicator of the association between keratin and intestines in human bodies, they are not exact replications. Thus, there will be differences seen between the two. For example, AOM-induced tumors in mice never express the p53 mutation and rarely express the *K-ras* mutation [37,40]. Therefore, even if the DDS- or AOM-induced tumors and human tumors resemble each other, the development of nonspontaneous tumors can always function and develop differently than spontaneous tumors [37].

Most of the time, the combination of DSS and AOM is used to fully induce the mice to colon injury [41]. This is commonly called the AOM/DSS model. It is carried out by injecting the mouse with AOM and then administrating the mice with DSS orally. The combination of the two allows for an inflammation-related CRC model [41]. This type of stress induction allows for the process of the tumor development within the mice to take much less time while also being an accurate model of human intestine diseases [42].

### 3.2. Keratin Association with Mouse Models of Intestinal Diseases

Several studies of intestinal phenotypes of keratin-associated mouse models are summarized in Table 2. K8 knockout (K8−/−) mice are frequently used in studies to study keratins and their association with inflammatory bowel disorder, because this phenotype is most likely equivalent to one of the inflammatory bowel disorders found in humans, ulcerative colitis [43]. K8−/− mice showed spontaneous chronic Th2-type colitis, which is the closest in resemblance to ulcerative colitis found in humans [44], whereas K7−/− mice showed no intestinal phenotypes under basal conditions [45]. Furthermore, the basal phenotype of the K8−/− mice showed more potential signs of colorectal cancer, such as inflammation, hyperplasia, and apoptosis resistance [33]. Meanwhile, both heterozygous knockout (K8+/−) and wild-type (K8+/+) mice did not show any signs of intestinal inflammation. Instead, K8+/− mice showed the susceptibility to experimental colitis and an increased crypt length, while K8+/+ simply showed an increased crypt length [33]. In addition, the treatment of K8−/− mice with anti-β4-integrin antibody, which activates integrin-mediated survival signals, resulted in the upregulation of survivin and phosphorylation of focal adhesion kinase with reduced activation of the caspases [46]. Thus, in contrast to the proapoptotic effect of the K8 mutation or the absence of K8 in hepatocytes, the absence of K8 confers microflora-dependent resistance to colonocyte apoptosis.

Even though CRC is not spontaneously developed in K8−/− mouse models, a study that utilized both AOM and ApcMin/+ CRC murine models showed that there were higher numbers of tumors in the distal colon of the K8−/− mice compared to the controls [34]. In a similar way, K8+/− mice that were induced with both DSS and AOM developed more tumors in the intestines compared to the controls [48]. The K8+/− mice treated with DSS or AOM/DSS showed an enhanced colonic permeability, which may lead to the alteration of microbiota in K8+/− gut [48]. Indeed, after the AOM/DSS treatment, fecal microbiota of K8+/− mice were different from that of control mice. K8+/− mice with CAC showed the increased *Firmicutes* and *Proteobacteria* and the decreased *Bacteroidetes* and *Verrucomicrobia* [48]. Notably, K8 was downregulated in mice colons under DSS-induced colitis and AOM/DSS-induced CAC and in patient colons with cancer [48]. Overall, in the colonic epithelia, what can be concluded about K8 is that the expressed levels of K8 do help to protect mice from external stress, as well as during recovery if stress is induced.

In addition, a previous study using nontransgenic FVB/n mice demonstrated altered keratins in intestinal stress responses [22]. In DSS-induced acute model of colitis, the K8 protein level was slightly decreased, whereas the stress-responsive K8 phosphorylation was increased. The other keratins, K7, K18, K19, and K20, showed a few changes after the DSS-induced chronic colon injury. K7 and K20 ended up having a wider crypt distribution, as well as an upregulation of their protein levels, but K18 and K19 maintained normality even with the chronic DDS treatment [22]. K18 and K19 were then downregulated when the acute DDS treatment was induced [22]. In general, the intestinal keratins adjusted their levels based on the various factors. For example, the age of the mice could alter the regulation of keratins. It can be seen that the intestinal keratins in older mice are strongly upregulated [22]. As a result, more tests need to be conducted with K7, K18, and K19 in murine models to accurately prove that the phenotypes were mostly normal, because not enough were performed in comparison to the K8-null mouse models.

Along with the other keratins in the intestine, K20 is expressed and its phosphorylation at serine 13 acts as a stress intestinal goblet cell marker [49]. The Arg80 residue of K20 was highly conserved in most keratins, and its mutation in skin keratins resulted in several skin diseases. In transgenic mice that overexpressed the mutant K20 R80H, the keratin filament networks were disrupted in the small intestine, while the transgenic mice that overexpressed wild-type K20 had normal keratin filament networks [11]. On the other hand, the keratin filaments of K18 R89C, equivalent R80H mutation in K20, remained undisrupted in the intestines from transgenic mice that overexpressed K18 R89C, possibly due to restricted expression of K18 and the presence of other keratins, such as K19 and K20, in the small and large intestines [47,50,51]. Taken together, when multiple keratins of the same type are present in one organ, the keratins can have functions that are redundant or complementary [11]. As an example, when the transgenic mice (wild-type K20, mutant K20, wild-type K18, or mutant K18) were intermixed and cross-bred, mutant 20 was rescued by wild-type K18, and mutant 18 was rescued by wild-type K20 [11]. Not only does this show that these two keratins have redundant functions in the intestinal organization of keratin filaments, but it demonstrates that K19 could possibly have similar redundant functions because of its usual presence with K18 and K20 in the intestines [11,51].

The use of transgenic mice induced with DSS is a very useful mouse model. With the knowledge gained from comparing data on the role of keratins in mouse intestines, the role of keratins in human intestines can be learned about. However, unlike the liver and pancreas, not many mouse models have been used to study the role of keratin mutants in intestine diseases. As a result, more experiments need to be conducted to find a better understanding of the key functions or associations keratins have in intestine diseases. Then, they could be useful for the improvement of therapy for patients suffering from intestine diseases.

## 4. Keratin Mutations Associated with Intestine Diseases

### 4.1. Keratins and IBD

The various disorders that cause chronic inflammation in the intestines are said to be types of inflammatory bowel disease (IBD). There are two main subtypes of IBD: ulcerative colitis (UC) and Crohn’s disease (CD). While UC only occurs in the rectum and colon, CD can occur anywhere in the gastrointestinal tract [52]. However, CD mainly occurs in the ileum. Studies regarding IBD patients, specifically in both UC and CD, have been conducted based on mutations in K8 and K19 because of the spontaneous development of Th2-type colitis in K8-null mice [53,54] and the significant presence of K8 and K19 in intestinal epithelial cells [8]. K8 variants were found in patients who suffer from IBD [14], but it is still unknown whether the presence of these variants cause or predispose patients to IBD. The few studies regarding the major K8 variants in IBD patients are summarized in Table 3.

Both UC and CDD have similar symptoms in which the patient suffers from diarrhea, fever, abdominal pain, and possible malnutrition. IBD affects over 1 million people in the United States alone and around 2.5 million people in Europe [53]. IBD cases are developed during early childhood or adolescence about 25% of the time in patients, while about 10–15% of cases are detected in patients over 60 years old [54]. The exact etiology for these diseases is still unknown. However, genetics, sex, environmental factors, and immune systems seem to play a big role. For example, CD is more commonly found in female adults past the pediatric stage than CD in male adults [54]. Currently, there is still no curative treatment that is available for use. Thus, in Table 3, we summarize studies performed regarding the roles of keratin in IBD patients, which may bring up potential novel therapies for this disease.

In Table 3, the *p*-value was not determined in the conducted studies, except for one which involved German patients who suffer from IBD [55]. The study showed that the association of the K8 variant, G62C, was not significant in IBD patients due its *p*-value being greater than 0.05, *p* = 0.46 [13,55]. In the summarized studies, it is seen that the calculated quotients for the number of carriers divided by the total number is way too small to have a significant value. Whether the cohort is based on just UC, CD, or IBD, the value shows that these mutations in K8 do not greatly impact the intestine diseases of humans, unlike what was seen in the mouse models.

On the other hand, the K19 frequency in human patients is bigger in value than K8. In Table 3, the variants in the K19 promoter are found to be around 34–47% of the UD, CD, and IBD cohorts [13]. However, similar to K8, K19 does not show significant associations to intestinal diseases in humans due to K19 variants being a common polymorphism [19]. With just one study screening the K19 gene in IBD patients, it is difficult to clearly state the associations between keratins and the diseases. More studies that involve K19 will need to be further investigated.

As a result, the common mutations in K8, such as Y54H, G62C, and R341H, and K19 demonstrate that they have no significant association with IBD [13,55]. Even if they might show some significance, the genetic backgrounds of the patients can alter the potential measurements of keratin variants [13]. Interestingly, a study performed in vitro illustrated the impact of the keratin mutations by generating colonocyte cell lines expressing keratin variants that have been identified in IBD patients [56]. The results showed that the mutations alter the intestinal cell barrier function, which suggests that the variants may contribute to producing the disease phenotype in vivo. Thus, more studies may need to be conducted with different measurements, such as in vitro versus in vivo, in order to further investigate keratin’s association with intestine diseases or to find other keratin variants that have a greater impact on the diseases.

Notably, keratin levels in the mucosa in IBD patients are altered in acute inflammation that predisposes to development colitis-associated cancer (CAC) [57]. Keratin levels in mucosa are decreased in acute inflammation, which may cause weakening of mucosal IF protein integrity, but the levels are restored or increased after clinical and endoscopic remission [57]. It remains to be determined whether the impaired keratin expression in repeated inflammation is associated with CAC progression. Interestingly, a recent study demonstrated that K1 was involved in maintaining the intestinal barrier in UC by the upregulation of tight junction proteins, such as occludin and ZO-1 [58].

### 4.2. Keratins and CRC

One of the most common cancers that Americans suffer from is colorectal cancer (CRC) [59]. In 2017, more than 135,000 new diagnoses of CRC and 50,000 deaths from this cancer were detected [59]. The majority of the new cases were found in patients older than 65. CRC is the development of cancer in parts of the large intestine. It is sometimes referred to as colon carcinoma, colon cancer, rectal cancer, or bowel cancer. Like many other cancers, CRC has various types. CRCs can be adenocarcinomas, carcinoids, lymphomas, and others. However, the most common of all these types is adenocarcinomas. Almost 90% of CRCs are considered to be adenocarcinomas, which is a type of CRC that originates from colorectal mucosa epithelial cells [60,61]. In a similar manner to IBD, the symptoms of CRC include blood in stool, abdominal pain, fatigue, and weight loss. While the main causes are due to aging and changes in one’s lifestyle, sometimes the cause is due to other genetic disorders within the human body [18,59]. In the beginning stages of CRC, the tumor starts off as a polyp, or a benign tumor, but as time progresses, it can become malignant [18]. Here, we summarize studies performed regarding the roles of keratin in CRC patients, which may bring up potential innovative therapies for this disease.

K7 and K20 are potential biomarkers in CRC [23,27]. While K20 is mostly found in the villus cells of the gastric and intestinal epithelium, K7 is mainly found in simple glandular epithelia [8,62]. Moreover, K20 is expressed in the normal colonic epithelium near the upper regions of the crypts, but K7 is expressed in colorectal carcinoma, mostly near the atrophic epithelium [62]. When studying CRC, it is common to use a combination of both K7 and K20 due to their increased diagnostic value, specifically in carcinomas [63]. Although the role of these keratins has not been fully identified within the intestines, studying these keratins might help determine the origin of primary and metastatic carcinomas, since different epithelial tissues express unique keratin patterns, and the patterns are frequently sustained in carcinomas and their metastases [23,27,64]. K20 is continually expressed in colorectal cancers, but different colorectal carcinomas may express different keratin patterns [62]. However, while some colorectal carcinomas simply express K20− or K7+, in general, the immunoprofile, K7−/K20+, is used as a multi-marker phenotype within various studies, because it is characteristic of CRCs [65,66]. Around 75–95% of CRC cases detected this multi-marker phenotype [67]. In addition, previous studies showed the mechanistic relationship between alterations in keratins and the cancer phenotype. For example, aberrant K7 expression is associated with poor tumor differentiation and tumor budding that could lead the tumor to further develop near adjacent stroma [26]. K7 expression is detected more often in advanced stages of CRC, and K20 overexpression is found in adenocarcinomas as compared to nonneoplastic colorectal epithelium [68].

Table 4 summarizes two studies conducted on the expression of K7 and K20 in CRCs, specifically primary CRCs, and colorectal carcinoma. In a study involving primary CRCs, 17.3% (34 out of 196) of the colorectal adenocarcinomas stained positively for K7, and 81.1% (159 out of 196) stained positively for K20 [63]. However, in a study that focused on colorectal carcinomas, K7 was expressed in 9.3% (21 out of 225), and K20 was expressed in 73.3% (165 out of 225) [62]. Even though the expression of K20+ was overall higher than K7+ in the two studies, the percentages of the expressions in different studies showed that various factors, such as the histologic grade, as well as the tumor location, affected the percentages [62,68]. As shown in Table 4, K7−/K20+ had the highest percentage, whereas K7+/20− had the lowest percentage in both studies [62,63]. However, depending on how advanced the cancer is, the multi-marker phenotype value could vary. In a study performed using tissue microarrays (TMAs), it was shown that the K7+/K20+ profile was expressed greatly in advanced cancers, while K7−/K20+ was expressed greatly in early-stage cancers [68]. In summary, K20+ was expressed higher in patients who suffered from CRC and can possibly be used as a clinical biomarker to determine the origination of different cancer cells.

K18 and K19 are also prognostic indicators of CRC. The overexpression of K18 is frequently observed in many types of human cancer [69]. In terms of CRC, a study using 108 colon cancer tissues demonstrated that the expression of K18 was indeed expressed in higher amounts in CRC as compared with normal tissues [70]. In addition, the increased level of circulating intact K18 (measured by M65 antibody) and caspase cleaved K18 (measured by M30 antibody) in serum was observed in the patients with CRC compared with the controls [71], and the high level of circulating K18 fragments was associated with systemic inflammation [72]. In the case of K19, the K19 fragment (referred to as CYFRA 21-1) was significantly increased in the serum of CRC patients [73], and K19 was utilized as one of the biomarkers for the early detection of CRC [74]. Interestingly, one study demonstrated a molecular mechanism between K19 and a cell signaling pathway in breast cancers [75]. When K19 was bound with the β-catenin/RAC1 complex, this resulted in an increase in the nuclear translocation of β-catenin and led to the enhancement of NUMB expression [75]. Thus, K19 can regulate the NUMB-dependent NOTCH signaling pathway in breast cancers [75]. This finding may help understand the molecular basis of K19 in other cancers, including CRC. Taken together, K18 and K19 are utilized as prognostic markers of CRC.

## 5. Conclusions

Analyses of keratin variants in patients with IBD provide insufficient evidence for keratin function in these diseases. It remains unclear whether the keratin mutations have significant effects on IBD, since the frequency of most mutations in K8 and K19 was not statistically significant. Although one in vitro study demonstrated that the keratin variant, such as K8 G62C found in IBD patients, altered the barrier function of intestinal cells [56], analyses of the intestinal phenotypes in transgenic mice expressing K8 G62C have not been reported yet. So far, most experiments shown an intestinal phenotype were performed by using K8-null mice. Research using transgenic mice that express K8 G62C or R341H/C will likely provide important biological and pathogenesis information regarding the importance of the keratin variants in intestinal diseases. In terms of the role of intestinal keratins as biomarkers for CRC, it is relatively well-studied. However, it is unknown whether the intestinal keratin mutations have an effect on CRC progression.

## Figures and Tables

**Table 1 ijms-23-08051-t001:** Distribution of keratins in intestine.

		K7	K8	K18	K19	K20	Ref.
Normal Intestine Cells	Human Small Intestine	(−)	+	(+)	+	(+)	[11,12]
	Human Large Intestine (Colon)	(−)	+	+	+	(+)	[12]
	Mice Small Intestine	(+)	+	(+)	+	(+)	[11]
	Mice Large Intestine (Colon)	(+)	+	+	+	(+)	[11]
Abnormal Intestine Cells	IBD	(−)	+ ^M^	(−)	(−)	(−)	[13,14]
	CRC	+ *	+	+	+	+	[15,16,17,18]

(−), negative; (+), weakly expressed; +, expressed; + *, expressed in some cases, + ^M^, expressed as a mutation; IBD, inflammatory bowel disease; CRC, colorectal cancer.

**Table 2 ijms-23-08051-t002:** Intestinal phenotypes of keratin-related mouse models.

Mouse Line	Intestinal Phenotype	Ref.
K7−/−	No disease phenotype on colon tissue histology	[45]
K8−/−	Residual K18 and K19 aggregates in proximity of cell surface in small and large intestine	[43]
	Colorectal hyperplasia, spontaneous Th2-type colitis	[43,44]
	Predisposition to colorectal cancer	[20,34]
K18 R89C	Minimal disruption of keratin filament in colon under basal condition, no effect on tissue histology	[47]
K20 R80H	Disrupted keratin filament in small intestinal villus under basal condition, no obvious effect on tissue histology	[11]

**Table 3 ijms-23-08051-t003:** Comparison of the keratin variant frequency in patients with IBD and the controls.

Variant	No. of Variant Carriers/Total (%)				*p*-Value	Ref.
	UC Cohort	CD Cohort	IBD Cohort (UC + CD)	Controls		
K8 G62C	8/348 (2.3)	6/555 (1.1)	13/866 ^a^ (1.5)	6/273 (2.2)	ND	[13]
	2/32 (6.3)	1/57 (1.8)	3/89 (3.4)	1/97 (1.0)	ND	[14]
	3/131 (2.3)	5/217 (2.3)	8/348 (2.3)	9/560 (1.6)	*p* > 0.05	[55]
K8 Y54H	0/131 (0.0)	0/217 (0.0)	0/348 (0.0)	0/560 (0.0)	ND	[55]
K8 R341H/C	14/348 (4.0)	24/555 (4.3)	37/866 ^a^ (4.3)	11/273 (4.0)	ND	[13]
K19 Promoter	34/100 (34.0)	37/100 (37.0)	71/200 (35.5)	33/70 (47.1)	ND	[13]

^a^ The numbers for IBD cohort are less than the sum of the UC + CD cohort, because each sibling in group II UC-CD sibling pair is counted in either UC or CD cohort, but only one of these siblings is counted in the IBD cohort [13]. ND, not determined.

**Table 4 ijms-23-08051-t004:** Keratin 7 and keratin 20 expression patterns in colorectal carcinoma.

Total No. of Samples	No. K7+ (%)	No. K20+ (%)	No. K7+/20− (%)	No. K7−/K20+ (%)	No. K7+/20+ (%)	No. K7−/20− (%)	Ref.
196	34 (17.3)	159 (81.1)	4 (2.0)	129 (65.8)	30 (15.3)	33 (16.8)	[63]
225	21 (9.3)	165 (73.3)	9 (4.0)	153 (68.0)	12 (5.3)	51 (22.7)	[62]

No, number; Keratin positive (+) or negative (−) in cells based on immunohistochemistry.

## Data Availability

Not applicable.

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
