# Peer review of "Roles of Keratins in Intestine"

_ijms, 2022, doi:10.3390/ijms23148051_

Round 1
Reviewer 1 Report
This manuscript provides an overview of keratin expression in the intestine and alterations of keratin expression related to intestinal pathologies, including inflammatory bowel disease and colorectal cancer. The manuscript uses many citations to support observed variations in keratin expression patterns. The compilation of data is useful and I think the manuscript does begin to show that keratins are more than just clinical biomarkers; keratins have cellular function that are important to maintenance of the intestine. As a result, changes in keratin gene expression patterns have direct implications to intestinal pathologies.
Unfortunately, the manuscript falls short in a couple of areas. It cites many differential changes in keratin gene expression, both normal and abnormal, but does not place them into a summary context. What is the significance of changes in a keratin’s gene expression? What is the significance of the specific different keratins expressed in the intestine? Why are changes in keratin gene expression relevant prognostic indicators and how might they directly affect the cell or tissue morphological character?
Additional points and suggestions are provided below for guidance to improve the manuscript.
Table 1: Use a (-) sign to denote negative rather than leaving it blank. Blank suggests that it has not been examined.
Line 63: A clear set of statements needs to be made articulating the relevant keratin heterodimer pairings. For instance, up to this point, no mention of K19-K8 dimerization has been made, leading one to believe that K7 is K19’s primary heterodimerization pairing, which is not accurate. Secondly, although many keratin pairings have been shown possible by way of in vitro assays, many of these have not been shown to exist in cells. One needs to use discretion in extrapolating these claims.
The description of the keratin pairing is really important for placing these proteins into their cellular context. Since the keratins form obligate heterodimers, changes in expression or regulation of one keratin has ramifications to its corresponding partner keratins. It seems to me that a good description of this is absolutely necessary to make any sense of subsequent discussion regarding differential keratin gene expression patterns.
Line 83: I am uncertain that such direct comparisons were made between K8, K18, and K19 in Ref 25. I’ve looked through this paper and I am unable to find supporting evidence of the statement made in the current manuscript. The closest I found was in Figure 2B that has a 2D gel of keratins in the colonic mucosa, which were microdissected from crypts. In this figure however the levels of K8, K18, and K19 look comparable. And I could not find any mention of keratin expression in villi in this paper.
Line 111: The section entitled ‘General Role of Keratins’ mostly seems to pertain to the general roles of keratins in cells rather than general roles of keratins in medical diagnostics. That keratins are used as biomarkers is a result of their specific expression in particular tissues under certain conditions for often presently unidentified reasons. While their selective expression is certainly curious, the usage of keratins as biomarkers does not indicate the important roles that keratins do play in cells. Moreover, the reference to non-small cell lung carcinoma seems out of place. Keratins certainly have general roles to cell biology, but their biomarker expression in NSCLC does not illustrate this point. I would like to see better examples of the general roles of keratin filaments in cells provided here, including specific mention of linkage to cell-cell and cell-matrix adhesions. These are particularly relevant given that intestinal pathologies affect tissue morphology.
Line 151: Section title would be more accurately written as ‘Keratin Association with Mouse Models of Intestinal Diseases’
Table 2: The descriptions of the “Intestinal Phenotypes” are not clear, as an example the K7 -/-. Are these descriptions intended to be of the keratin filament organization in intestinal cells or the more general tissue morphology/pathology of the intestine? It currently seems to be a mix of both. Some of the descriptions discuss keratin filaments, but the apparent consequence to the more general intestinal phenotype is not consistently stated. This makes for a confusing comparison between keratin knockouts and mutations. In addition, the type of ‘disrupted keratin filaments’ could be better explained. Otherwise, one is led to believe these disruptions are all equivalent.
Lines 213-264: The authors attempt to tie keratins to IBD, although the evidence linking them seems weak. Indeed, given the protective strengthening that keratin provides to cells and tissues, one would expect keratin knockout animals to be more susceptible to IBD and Crohn’s than has been demonstrated. The authors point to minor (but statistically insignificant) trends in the data and make the appeal to do more studies. I suggest reorganizing this section to first present the role(s) that keratins do play that would suggest association with IBD (e.g. lines 229-231 (refs 8, 53, 54, 24, 56). Then, discuss the clinical data and the shortcomings of these studies. Then the authors can make the case that further investigations are warranted.
Lines 287-288: “Studying these keratins might lead to an understanding of the origination of carcinomas… the added underlined part significantly changes the meaning of this statement.
Section 4.2 focuses exclusively on K7 and K20 even though Line 279 states, “Here, we summarize studies performed regarding the roles of keratin in CRC patients…” Is there a reason for this narrow focus on these two keratins? There are quite a few papers that discuss using K18 and K19 as prognostic indicators of cancers including CRC. The exclusion of these keratins from this section is both surprising and odd given the literature. This section of the review should also probably go a bit further and not only talk about keratin expression patterns but also the mechanistic relationship between alterations in keratins and cancer phenotype.
The Conclusion circles back to exclusively IBD with no mention of CRC or other keratin-related pathologies, and the case for IBD being a keratin-related disease was not well-supported. If the authors wish to make the case that IBD could be a keratin-related disease, then evidence should be stated demonstrating the important roles that keratin is known to play. As noted for the IBD section within the review manuscript, use this as starting point to justify that further investigation of keratin-related intestinal pathologies are needed.
Reviewer 2 Report
The manuscript by Mun et al is quite well written and already quite comprehensive. Nevertheless, I noticed a few things in the review that can/should be improved.
Just Keratin levels are reduced in acute enteritis but increased after clinical and endoscopic endoscopic remission. There is a publication (Corfe et al. BMJ Open Gastro, 2015;2) showing that keratin concentrations are slightly higher than normal in patients with persistent disease, but in active inflammation and in inflammation of patients with recent disease have a reduced keratin concentration is seen. Taken together, these data suggest a delay in the recovery of keratins in the mucosa after acute inflammation despite mucosal healing.
Furthermore, the gut mechanical barrier plays a key role in the pathogenesis of ulcerative colitis. A study (Wu et al. Genes Genomics 2021 43:1389) showed that keratin1 was downregulated in UC, but the mechanism by which KRT1 affects the gut barrier is unknown. Keratin1 maintains the gut barrier by upregulating Tight Junction proteins in UC. The expression of occludin and ZO-1 decreased after DSS induction, and the decrease in occludin and ZO-1 was smaller in the krt1-TG group than in the WT group, which increased significantly after MP, while the expression of claudin-2 showed the opposite effect. This could be mentioned.
The authors have already described the phentype of K8 KO mice in their manucript, but treatment of the K8 KO mice with an anti-β4 integrin antibody resulted in an interesting effect: upregulation of survivin and phosphorylation of focal adhesion kinase with reduced activation of caspases. (Habtezion et al. 2011, PNAS 108;4) Thus, in contrast to the proapoptotic effect of the K8 mutation or the absence of K8 in hepatocytes, the absence of K8 confers microflora-dependent resistance to colonocyte apoptosis.
In another study by Lui et al. (Oncotarget, 2017, Vol.8,No57), they found that K8 is downregulated in the colon during DSS-induced colitis and AOM/DSS-induced colitis-associated colorectal cancer (CAC) development. In human patients with colorectal cancer, CK8 is downregulated. As colon permeability is increased by inflammatory treatment with DSS or AOM/DSS, this may also lead to alteration of the gut microbiota in K8 KO mice with CAC. Liu et al. showed that Firmicutes and Proteobacteria are increased in K8 KO mice with CAC, while Bacteroidetes and Verrucomicrobia are decreased. Here, a small chapter might be worth adding.
In my opinion, these 4 additions would round off the article and give the reader a comprehensive overview.
Round 2
Reviewer 1 Report
The authors have satisfactorily addressed most of my concerns. There is one issue in the revision that I feel will require further revision.
I disagree that data for K18 and K19 as prognostic indicators of CRC is “limited” in comparison to K7 and K20, as the authors state in their revised manuscript (Lines 360-361). In addition to the four articles cited in the manuscript, there are many other papers supporting K18 and K19 involvement. Clinically, K19 is one of the stronger correlative markers of metastatic CRC and has even been shown recently to have some early prognostic value. A few examples below are in addition to that which the authors have already cited (and there are many more). The authors should decide whether in this manuscript they want to strengthen the support of K18 and K19 in this section or remove the wording that inaccurately lessens the data on K18/K19.
Bhardwaj, M., Weigl, K., Tikk, K., Benner, A., Schrotz-King, P. and Brenner, H. (2020), Multiplex screening of 275 plasma protein biomarkers to identify a signature for early detection of colorectal cancer. Mol Oncol, 14: 8-21. https://doi.org/10.1002/1878-0261.12591
Mostert B, Sieuwerts AM, Bolt-de Vries J, Kraan J, Lalmahomed Z, van Galen A, van der Spoel P, de Weerd V, Ramírez-Moreno R, Smid M, Verhoef C, IJzermans JN, Gratama JW, Sleijfer S, Foekens JA, Martens JW. mRNA expression profiles in circulating tumor cells of metastatic colorectal cancer patients. Mol Oncol. 2015 Apr;9(4):920-32. doi: 10.1016/j.molonc.2015.01.001
Saha, S., Choi, H., Kim, B. et al. KRT19 directly interacts with β-catenin/RAC1 complex to regulate NUMB-dependent NOTCH signaling pathway and breast cancer properties. Oncogene 36, 332–349 (2017). https://doi.org/10.1038/onc.2016.221
Päivi Sirniö, Juha P. Väyrynen, Shivaprakash J. Mutt, Karl-Heinz Herzig, Jaroslaw Walkowiak, Kai Klintrup, Jyrki Mäkelä, Tuomo J. Karttunen, Markus J. Mäkinen & Anne Tuomisto (2020) Systemic inflammation is associated with circulating cell death released keratin 18 fragments in colorectal cancer, OncoImmunology, 9:1, DOI: 10.1080/2162402X.2020.1783046
